# Can analyst coverage reduce corporate tax avoidance? Evidence from China

Xiaofei Shi[1,2], Yuanhao Shen[1,2], Yuanfang Wang[3*], Jinlong Han[1,2]

1 School of Business Administration, Hebei University of Economics and Business, Shijiazhuang, China,
2 Research Center for Corporate Governance and Enterprise Growth, Hebei University of Economics and Business, Shijiazhuang, China, 3 School of Accounting, Capital University of Economics and Business, Beijing, China

* wangyuanfang_cueb@126.com

## Abstract

Tax avoidance is a widespread problem, much explored in the literature. Using a sample of A-share listed companies in China from 2009 to 2021, this study finds that analyst coverage significantly inhibits corporate tax avoidance behavior, mainly by improving the information environment and alleviating agency problems. Further analysis finds that the role of analyst coverage is more significant in firms where investment in innovation is limited, state-owned enterprises, and those with low management shareholding. The paper enriches the relevant literatures about analyst coverage and corporate tax avoidance, identifies the potential to inhibit corporate tax avoidance from the perspective of information environment and agency costs and provides suggestions for regulators and corporate governance.

## 1. Introduction

Fiscal revenue plays an important role in the macro-control of national economic development, infrastructure construction, and investment in projects related to the national economy and people's livelihood. According to China's National Bureau of Statistics, tax revenue accounted for 90% of Chinese fiscal revenue from 1990 to 2018. Gradual reform over 40 years has improved and developed China's tax system, however, further improvements are required to taxation policies and tax collection and management systems and in the professional capabilities and quality of tax personnel [1]. Corporate income tax in China accounts for a quarter of pre-tax profits, which motivates managers to avoid tax, facilitated by imperfect tax laws that allow companies to make use of loopholes in the collection and management system. Given this behavior is not consistent with the intent of China's tax legislation, it has prompted significant research attention.

Existing studies focus on two aspects: company characteristics and the external environment. Company characteristics include political connections, the nature of property rights [2], personal characteristics of management [3], management

**Data availability statement:** All relevant data are within the manuscript.

**Funding:** This work was supported by the Key Project of Social Science Research of Beijing Municipal Education Commission (SZ202210038021/21GJB021), New Finance and Economics Research Project of Hebei University of Business and Economics (2020XCJ03). The key project of social science research of Beijing Municipal Education Commission (SZ202210038021/21GJB021) is led by Teacher Wang Yuanfang, Wang Yuanfang.conducted the empirical analysis and finalized the manuscript. The new finance research project of Hebei University of Economics and Business (2020XCJ03) is led by Shi Xiaofei. Shi Xiaofei designed the paper, collected the data, summarized the literature review, developed the hypotheses.

capabilities and powers [4], corporate reputation [5], or social trust [6] and so on. The external environment includes policy uncertainty in the region where the company is located, the level of intensity of tax collection and administration [7], surveys by institutional investors [8], the geographic distance between the company and the relevant regulatory authority [9], and so on. Few studies consider the possible impact of capital market factors, such as analysts.

Yet, as important participants in the capital market, analysts are increasingly valued by investors. Analysts provide investors with relevant information by ongoing analysis of a firm's performance, which is published in research reports. They act as "information intermediaries", reducing the degree of information asymmetry between stakeholders and enterprises [10]. They also have a "supervisory effect" in observing the decisions and behaviors of managers, in particular, opportunistic behaviors such as earnings management [11]. However, some studies find that the media and analysts, as an important part of the external governance mechanism, exert a "pressure effect" [12], that is, managers act to cater to analysts' forecasts. Studies find that analysts tend to be optimistic about the earnings forecasts of companies based on certain interest relationships [13,14]. If the company's earning is less, then its stock prices may decline, and managers may face salary reduction or dismissal. Hence, some managers may engage in earnings management activities, such as tax avoidance, to meet analysts' forecasts.

Logically, analyst coverage can promote or inhibit corporate tax avoidance. On the one hand, analyst coverage subjects a company to a greater degree of external supervision, hence the policies and actions taken by management will receive more attention. The risk of self-interested behavior being discovered increases, and the possibility of failure of tax evasion activities also increases. The cost to be paid is far greater than its benefits and therefore analyst coverage will inhibit management's tax avoidance. On the other hand, analysts' relatively optimistic forecasts spark higher expectations of the company's performance, increasing pressure on management. To mitigate risks associated with failing to meet expectations, managers are motivated to find other ways to meet analysts' forecasts, for example, by adopting tax avoidance.

Using a sample of Chinese A-share listed companies from 2009 to 2021, this study finds that analysts' attention reduces corporate tax avoidance behavior; that is, it has a "supervisory effect" on management. The path test employed shows that the effect of analysts' coverage is mainly achieved by improving the information environment and alleviating agency problems. Other tests show that corporate innovation investment weakens the inhibiting effect of analysts' attention on corporate tax avoidance; the "supervision effect" is more pronounced in state-owned enterprises; and the monitoring effect of analyst attention is more significant in companies with lower management ownership than in companies with higher management ownership.

This study contributes to the literature by advancing studies on tax avoidance from an analyst perspective. External environment factors that influence corporate tax avoidance are mainly concentrated in the tax collection and management intensity of the company's location, policy uncertainty, financial development level, media reports, political connections, and so on. Less consideration is given to the possible

impact of analysts on corporate tax avoidance. The study also provides evidence of the supervisory effect of analyst coverage and broadens the research perspective of analyst coverage on the influence of tax avoidance behavior, given that the influence of corporate tax avoidance is rarely investigated from the perspective of information environment and agency costs. By conducting a path test we verify the mechanism underpinning the influence of analyst's coverage on tax avoidance behavior.

The rest of the paper is organized as follows: Reviews the literature and develops hypotheses. Details the method used. Present the empirical results and analysis. Discusses the findings and the final concludes with implications for research and policymaking.

## 2. Literature review, theoretical analysis, and research hypothesis

### 2.1. Literature Review

**2.1.1. Analyst coverage.** The analysts play and the degree of influence of this role in a company's external governance mechanism has been widely studied. Some studies find that analysts play a beneficial role. Zhu, He, and Tao [15] find that analysts' information gathering activities can increase the information content of stock prices and improve the operational efficiency of the capital market. Zhang and Zhou [16] find that analysts' coverage can effectively reduce the accrued earnings management and true earnings management of listed companies and enhance the transparency of company information. Pan, Dai, and Lin [17] investigate the internal relationship between company information transparency and the risk of falls in individual stocks, finding that a country's weak institutional environment is an incentive for analysts to reduce information opacity. Kim et al. [18] proves that analysts are important information intermediaries and monitors and, thus, that analyst coverage influences the underlying stock's expected crash risk. Chan et al. [19] finds that, analyst coverage reduces firms' cost of capital and thereby facilitates more investments in organization capital. Their findings contrast with other studies that have shown how the adverse effect of analyst coverage enhances managerial myopia and reduces corporate R&D.

However, some studies identify negative effects from analysts' coverage. O'Brien, McNichols, and Hsiou [20] find that correlation analysts are motivated to respond to company's good news in a timely manner, rather than releasing bad news. Song and Zhang [21] find that, compared with non-affiliated analysts, affiliated analysts are more likely to overestimate company earnings, indicating that affiliated analysts are not independent and more likely to make earnings forecasts that are beneficial to listed companies. Xu et al. [22] find that when analysts face a "conflict of interest", their optimism bias will increase the impact of the risk of stock price collapse. Zhou and Zhao [23] find, in their study of the characteristics of analysts' optimistic earnings forecasts, that the interest linkages between securities companies and listed companies and the heuristic cognitive biases of the analysts themselves make it easier for analysts to issue optimistic forecast reports.

Based on the above, it seems the role of analysts in Chinese capital market is gradually increasing and that analysts may reduce information asymmetry and inhibit company's earnings management because of their independence, or they may lose their independence because they are tied to the company's interests.

**2.1.2. Corporate tax avoidance.** An increasing number of studies investigate tax avoidance as its prevalence increases worldwide. From the perspective of influencing factors, Li and Xu [24] find that companies with political status engage in more tax avoidance behaviors, and when local governments face greater economic growth and fiscal pressure, the tax avoidance effect of political status is more obvious. Chen and Xu [25] find that high-quality internal control can help companies achieve their compliance goals, reduce their increased risk of violations due to aggressive tax avoidance, and protect the interests of investors. Chen, Chen, and Dong [7] examine the impact of policy uncertainty on corporate tax avoidance, finding that it increases tax avoidance, especially in regions with low tax collection and management. Cao et al. [26] find that firms with higher customer concentration are more likely to engage in tax avoidance, easing financing constraints is one way to reduce customer concentration's influence on tax avoidance. Hjelström et al. [27] find that CEOs' and CFOs' personal tax behavior is related both to nonconforming and conforming corporate tax avoidance.

Liu and Ye [28] study the economic consequences of tax avoidance, finding an impact on investment efficiency, and that tax avoidance activities aggravate information asymmetry inside and outside the company, triggering excessive investment by enterprises. Ye and Liu [29] find that the higher the tax avoidance of listed companies, the higher the internal agency costs, and this effect is partly due to its reduced sensitivity to executive compensation and corporate performance. Cheng, Li, and Zheng [30] examine the impact of corporate tax avoidance under different monetary policies, finding that when monetary policies are loose corporate value is reduced, and when they are tight monetary corporate value increases.

Compared with foreign developed markets, the focus of Chinese analysts is more complex. In developed markets, where corporate governance mechanisms are more mature and investor protection laws are well-established, analysts are likely to play a more intermediary role in influencing market expectations and stock prices by providing professional analysis. In China, analysts not only act as information intermediaries, but also assume the role of external governance mechanisms to a certain extent, and their oversight effect plays an important role in curbing corporate tax evasion and other misconduct. At the same time, the peculiarities of the Chinese market also make analysts' optimistic forecasts and other behaviors may exert more pressure on management and affect their decision-making. Therefore, the influence mechanism and effect that analysts focus on in China's institutional environment are significantly different from those in foreign developed markets.

As the role of analysts in the capital market gradually increases, the impact on corporate tax avoidance will also increase. This paper contributes to the literature reviewed above to further examine the possible relationship between analysts' coverage and corporate tax avoidance.

## 2.2. Theoretical analysis and research hypothesis

Two opposing views exist on the impact of tax avoidance on enterprises. The traditional view considers that there is no agency cost between shareholders and management, and tax avoidance supports a company's cash flow, providing more cash for management to invest in projects that increase the value of the company, in turn having a positive impact on the company [31]. However, the opposing view based on principal–agent theory posits that the interests of shareholders and management are inconsistent, and managers are more likely to put their own interests above those of shareholders and the company. Tax avoidance behavior usually requires complex planning, which strengthens the degree of information asymmetry between shareholders and management. Managers' self-interest means that they are more likely to use their information advantage to invest cash resulting from tax avoidance for on-the-job consumption [30].

However, there is no consensus in the literature as to the impact of analysts on corporate tax avoidance behavior [25]. First, some scholars believe that analysts' coverage will reduce corporate tax avoidance because when analysts pay attention to a company, their evaluations are published in research reports that help investors better understand the company and reduce the degree of information asymmetry [32]. Moreover, because analysts have rich professional knowledge and industry experience, they have the expertise to discover earnings management behaviors such as corporate tax avoidance activities [32], so analyst coverage has a "supervisory effect". Second, investors have a high degree of trust in analysts' research reports [33]. If the research report includes negative news about corporate tax avoidance behavior investors' expectations of the company are lowered, leading to a decline in stock prices and damaging the company's value and reputation [34]. It will also attract the attention of tax regulatory authorities, and possible penalties may ensue. Therefore, when managers plan tax avoidance activities, they must consider the costs and benefits. Doing so suggests the following hypothesis:

**H1a: When analysts' concerns increase, the degree of corporate tax avoidance decreases.**

However, some studies find that analysts' attention exerts a "pressure effect" [35], prompting companies to avoid taxation. This is because analysts tend to be optimistic in their earnings forecasts. The study found a widespread optimism

bias in earnings forecasts made by analysts in China [36]. The reason for this may be that analysts are subject to certain conflicts of interest [37,20], with the result that companies are set high targets that are difficult to meet. Also, Xu et al. [22] find that analysts' optimism bias is significantly positively correlated with the company's future stock price collapse risk. This creates pressure for the management of the company – if the relevant expected value is not reached, it will have an impact on the company's stock price and further affect the interests of the managers themselves, including possible salary reductions or even dismissal. Therefore, in order to cater to the optimistic forecasts of analysts and to keep their own interests unaffected, the incentive for management to engage in tax avoidance behaviors increases. We propose the following hypothesis:

**H1b: When analysts' attention increases, enterprises' degree of tax avoidance rises.**

Our theoretical model is shown in Figure 1.

## 3. Research design

### 3.1. Sample selection

As China's corporate income tax reform began in 2008, this study collects the data of A-share listed companies from 2009 to 2021, conducting the following screening to reach a final sample: (1) exclude ST, PT, financial insurance listed companies; (2) exclude abnormal samples with actual tax rates less than 0 or greater than 1; (3) exclude samples with income tax expenses less than 0 and total pre-tax profits less than 0. When corporate profits are negative, the actual income tax rate calculated will be biased so we winsorized all continuousvariables at 1% and 99%. Standard errors are clustered at the firm level to control for heteroskedasticity and serial correlation among observations of the same firm. Finally, 8613 observations were obtained.

The nominal corporate income tax rate data comes from the Wind database, the management shareholding ratio and the actual controller data come from the CCER database, and the other data comes from the CSMAR database.

### 3.2. Variable construction

#### 3.2.1. Tax avoidance (TA).
The existing literature generally uses two methods to measure tax avoidance. First, measure from the perspective of the actual tax rate of the enterprise, second, use the accounting-tax difference of the enterprise to measure the degree of tax avoidance. Drawing on the research of [28], this study uses accounting-tax difference indicator BTD and the difference between nominal tax rate and actual tax rate RATE1 to measure the degree of corporate tax avoidance. The specific calculation method is shown in Table 1.

#### 3.2.2. Analyst's coverage (Analyst).
Take the natural logarithm of the analyst tracking number plus 1.

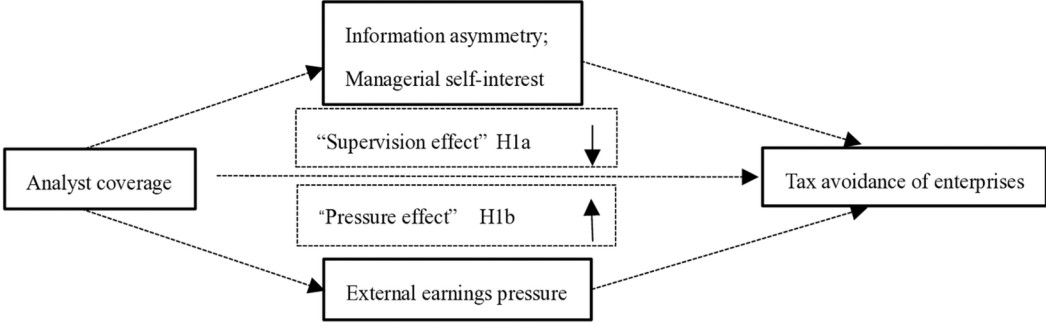

**Fig 1. Logical framework.**

**Table 1. Variable definitions.**

| Type | Variable | Description | Definition |
|---|---|---|---|
| Dependent variable | BTD | Accounting – tax differences | The difference between total accounting profit and taxable income. The calculation formula is: tax difference = (accounting profit before tax – taxable income)/total assets of the previous year; Taxable income = (income tax expense – deferred income tax expense)/nominal income tax rate |
| | RATE1 | Nominal tax rate – effective tax rate difference | (Income tax expense – deferred income tax expense)/EBIT |
| Argument | Analyst | Analyst coverage | Add one to the number of analysts tracking and take the natural logarithm |
| Mediation variables | Agencycost | Agency costs | Operating income for the period) / total assets at the end of the period |
| Adjust variables | R&D | Invest in innovation | R&D expenses/operating income |
| | SOE | Nature of property rights | The value of state-owned enterprises is 1, otherwise it is 0 |
| | EHR | Management shareholdings | At the end of the year, the total number of shares held by all senior management personnel of the company, except for directors and supervisors, accounted for the proportion of the total share capital |
| | ROE | Return on equity | Ending net profit/net assets |
| Control variables | ROI | Return on investment | Total investment income/investment assets at period end |
| | Growth | Company growth | Growth rate of operating income |
| | PPE | Fixed asset ratio | Closing fixed assets/Total assets |
| | Intang | Intangible asset ratio | Closing intangible assets/Total assets |
| | Lev | Asset–liability ratio | Total liabilities/Total assets |
| | INV | Inventory density | Inventory/Total assets |
| | LnSize | Company size | The natural log of total assets |
| | Year | Year | Annual dummy variable |
| | Indcd | Industry | Industry dummy variable |
| | Stock | Stock code | Firm dummy variable |

### 3.3. Empirical Model

We estimate the following model:

$$TA_{i,t} = \alpha_0 + \alpha_1 Analyst_{i,t} + \alpha_2 ROE_{i,t} + \alpha_3 ROI_{i,t} + \alpha_4 Growth_{i,t} + \alpha_5 PPE_{i,t} + \alpha_6 Intang_{i,t}$$
$$+ \alpha_7 Lev_{i,t} + \alpha_8 INV_{i,t} + \alpha_9 LnSize_{i,t} + Year + Indcd + Stock + \xi_{i,t} \tag{1}$$

where the explained variable $TA_{i,t}$ represents the degree of corporate tax avoidance, and the explanatory variable $Analyst_{i,t}$ is the number of analysts tracked by adding 1 to the natural logarithm. The remaining variables are defined in Table 1.

## 4. Empirical results and analysis

### 4.1. Summary statistics

Table 2 reports descriptive statistics for the main variables used in our study. The mean value of RATE1 is 0.009, indicating that the average actual tax rate of enterprises is lower than their nominal tax rate, and tax avoidance is widespread. The minimum value is −0.345, and the effective tax rate of some enterprises is much lower than the nominal tax rate, and there may be more aggressive tax avoidance behavior. The mean and median of BTD are also negative, indicating that more companies have accounting income lower than taxable income. It shows that the accounting profit of some enterprises is much lower than the taxable income, and the tax avoidance behavior is more obvious. The mean value of *Analyst* is 2.18, and the 1/4 quantile is 1.38, indicating that each company in the sample is followed by an average of two analysts. It shows that analysts pay more attention to listed companies. The standard deviation is 0.870, indicating that analyst attention varies between companies.

**Table 2. Summary statistics.**

| Variables | N | Mean | Std | Min | Median | Max |
|---|---|---|---|---|---|---|
| RATE1 | 8,613 | 0.009 | 0.092 | −0.345 | 0.008 | 0.249 |
| BTD | 8,613 | −0.004 | 0.043 | −0.276 | −0.004 | 0.259 |
| Analyst | 8,613 | 2.183 | 0.870 | 0.693 | 2.197 | 4.331 |
| ROE | 8,613 | 0.113 | 0.075 | 0.005 | 0.098 | 0.399 |
| ROI | 8,613 | 0.459 | 1.942 | −0.828 | 0.071 | 17.770 |
| Growth | 8,613 | 0.239 | 0.469 | −0.456 | 0.148 | 3.936 |
| PPE | 8,613 | 0.219 | 0.164 | 0.002 | 0.182 | 0.716 |
| Intang | 8,613 | 0.045 | 0.047 | 0.000 | 0.034 | 0.302 |
| Lev | 8,613 | 0.440 | 0.200 | 0.047 | 0.435 | 0.883 |
| INV | 8,613 | 0.161 | 0.151 | 0.000 | 0.123 | 0.750 |
| LnSize | 8,613 | 22.470 | 1.307 | 19.570 | 22.270 | 26.330 |

Table 3 presents the correlations of the main variables used in our study. *Analyst* has a significant negative correlation with *RATE1*, and a significant negative correlation with *Agency* cost. The largest absolute value of the correlation coefficient between the selected control variables is that the correlation coefficient between the asset-liability ratio *Lev* and the company size *LnSize* is 0.582, which is lower than 0.8, and the correlation coefficients among the remaining control variables are all lower than 0.5. The variance expansion factor VIF of each variable is lower than 10, so there is no multi-collinearity problem among independent variables.

## 4.2. The effect of analyst coverage on corporate tax avoidance

Table 4 presents the regression results of the effect of analyst coverage (*Analyst*) on corporate tax avoidance (*BTD, RATE1*). The coefficient of the explanatory variable (*Analyst*) is significantly negative at the 1% level, suggesting that analysts' concerns have inhibited corporate tax avoidance and exerted a "supervisory effect", supporting H1a. The main reason is that analysts focus reduces information asymmetry, hence corporate tax avoidance behaviors are more likely to be detected. In addition, the relevant decision-making behavior of managers has received more attention and external supervision. When planning for tax avoidance, the risk of failure increases, and its marginal income decreases. Therefore, based on the cost–benefit principle, managers will actively reduce tax avoidance behavior.

**Table 3. Analysis of correlation coefficients among main variables.**

| | BTD | RATE1 | Analyst | ROE | ROI | Growth | PPE | Intang | Lev | INV | Lnsize |
|---|---|---|---|---|---|---|---|---|---|---|---|
| BTD | | 0.813*** | 0.045*** | 0.163*** | 0.085*** | −0.023** | 0.087*** | −0.017 | −0.114*** | −0.095*** | −0.029*** |
| RATE1 | 0.033*** | | −0.025*** | 0.058*** | 0.049*** | −0.024** | 0.232*** | 0.009 | 0.172*** | −0.074*** | 0.117*** |
| Analyst | −0.002 | 0.003 | | 0.451*** | 0.021** | 0.173*** | −0.009 | 0.040*** | −0.019* | −0.037*** | 0.267*** |
| ROE | 0 | 0.102*** | 0.403*** | | 0.133*** | 0.260*** | −0.131*** | −0.060*** | 0.060*** | 0.003 | 0.132*** |
| ROI | −0.002 | −0.008 | −0.024** | 0.065*** | | −0.053*** | −0.054*** | −0.031*** | −0.015 | 0.019* | 0.116*** |
| Growth | −0.025*** | −0.013 | 0.034*** | 0.208*** | −0.009 | | −0.127 | −0.039*** | 0.026** | 0.005 | −0.024** |
| PPE | 0.002 | 0.241*** | −0.017* | −0.118*** | −0.039*** | −0.090*** | | 0.296*** | −0.006 | −0.290*** | 0.024** |
| Intang | 0.009 | 0.035*** | −0.005 | −0.032*** | 0.004 | −0.018* | 0.097*** | | −0.149*** | −0.189*** | −0.098*** |
| Lev | −0.008 | 0.177*** | −0.009 | 0.102*** | −0.036*** | 0.073*** | 0.057*** | −0.040*** | | 0.306*** | 0.567*** |
| INV | 0.001 | −0.077 | −0.051*** | 0.047*** | 0.021** | 0.036*** | −0.348*** | −0.231*** | 0.398*** | | 0.084*** |
| Lnsize | −0.007 | 0.121*** | 0.284*** | 0.150*** | −0.038*** | 0.014 | 0.088*** | −0.023** | 0.582*** | 0.191*** | |

Note: The lower triangle is Pearson's coefficient, and the upper triangle is Spearman's coefficient. *, ** and *** are significant at the level of 10%, 5% and 1% respectively.

**Table 4. Analyst coverage and Corporate Tax Avoidance.**

| Variable | (1) | (2) |
|---|---|---|
| | BTD | RATE1 |
| Analyst | −0.004*** | −0.007*** |
| | (−4.66) | (−4.08) |
| ROE | 0.114*** | 0.177*** |
| | (7.35) | (8.15) |
| ROI | −0.000 | −0.000 |
| | (−1.41) | (−0.70) |
| Growth | −0.008*** | −0.006** |
| | (−2.66) | (−2.16) |
| PPE | 0.017*** | 0.078*** |
| | (3.36) | (6.74) |
| Intang | −0.007 | 0.022 |
| | (−0.54) | (0.64) |
| Lev | −0.035*** | 0.061*** |
| | (−7.26) | (6.03) |
| INV | 0.004 | −0.021 |
| | (0.55) | (−1.23) |
| LnSize | 0.003*** | 0.003* |
| | (3.05) | (1.92) |
| cons | 0.008 | −0.007*** |
| | (0.38) | (−4.08) |
| Industry | Yes | Yes |
| Year | Yes | Yes |
| Stock | Yes | Yes |
| N | 8,613 | 8,613 |
| R² | 0.102 | 0.183 |

Note: The P value in parentheses, *, **, *** indicate significant at the level of 10%, 5%, and 1%, respectively.

In the regression results of the control variables, it can be seen that the regression coefficient of *RO*E is significantly positive at the 1% level, indicating that companies with strong profitability have a relatively high degree of tax avoidance. The fixed asset ratio is significantly positively correlated at the 1% level, because the more fixed assets a company has, the stronger its ability to avoid taxation through relevant policies, while the intangible asset ratio has no significant impact. There is a significant positive correlation between enterprise size and tax avoidance, indicating that large-scale enterprises are more inclined to avoid taxation.

### 4.3. Impact path test

According to the theoretical framework, as shown in Fig 1, tax avoidance, as a business strategy of a company, is mainly affected by the decision-making of managers, and its most direct effect is to reduce the company's cash outflow. However, due to factors such as separation of powers and insufficient salary incentives, managers' self-interest and on-the-job consumption motivates them to avoid taxation. This paper argues that analyst coverage suppresses corporate tax avoidance by improving corporate information transparency, supervising managers, reducing their self-interested behavior and on-the-job consumption, exerting a "supervisory effect". That is, analyst coverage can improve corporate information transparency, reduce agency costs, and then curb corporate tax avoidance. So, on the one hand, we use the modified Jones model to calculate the absolute value of manipulative accruals (*DACC*) as a measure of information transparency.

The greater the absolute value of manipulative accruals, the greater the probability of earnings manipulation and the lower the information transparency will be. On the other hand, the agent cost (*Agency cost*) is used to test whether the analyst coverage can affect the corporate tax avoidance by alleviating the agency problem.

Drawing lessons from the intermediary effect test methods of Wen et al. [38], Cheng, Tan, and Liu [39], and Hua, Liu, and Xu [40], the following models (2) and (3) were constructed.

$$DACC/Agencycost_{i,t} = \beta_0 + \beta_1 Analyst_{i,t} + \beta_2 ROE_{i,t} + \beta_3 ROI_{i,t} + \beta_4 Growth_{i,t} + \beta_5 PPE_{i,t}$$
$$+ \beta_6 Intang_{i,t} + \beta_7 Lev_{i,t} + \beta_8 INV_{i,t} + \beta_9 Lnsize_{i,t} + Year + Indcd + Stock + \xi_{i,t} \quad (2)$$

$$TA_{i,t} = \theta_0 + \theta_1 Analyst + \theta_2 DACC/Agencycost + \theta_3 ROE_{i,t} + \theta_4 ROI_{i,t} + \theta_5 Growth_{i,t} + \theta_6 PPE_{i,t}$$
$$+ \theta_7 Intang_{i,t} + \theta_8 Lev_{i,t} + \theta_9 INV_{i,t} + \theta_{10} LnSize_{i,t} + Year + Indcd + Stock + \xi_{i,t} \quad (3)$$

Following Luo [41], we use total asset turnover rate to measure agency cost. The calculation method is the current operating income/the total assets at the end of the period. The turnover rate of total assets reflects the degree of management's efforts; the lower the total asset turnover rate the more management is inclined to enjoy leisure time rather than hard work, which means a higher the agency cost between shareholders and management. The higher the total asset turnover rate, the lower the agency cost. To make comparison straightforward, we use the opposite of the total asset turnover rate to measure agency costs. The control variables are consistent with Model (1).

Table 5 presents the regression results of models (2) and (3). Columns (1)-(3) show the regression results with information transparency (DACC) as the mediating variable. In column (1), the coefficient of analyst attention (Analyst) and information transparency (DACC) is negative and significant. This indicates a significant negative correlation between analyst attention and information transparency. Specifically, when the attention of analysts towards a company increases, the company's information transparency significantly improves. Analysts, by continuously monitoring the company and releasing research reports, convey detailed information about the company to the market, including financial status, operating performance, and prospects. This information dissemination makes the company's information more transparent and reduces the information asymmetry between shareholders and management. The improvement in information transparency means that shareholders can better understand the actual operation of the company, thereby better supervising the behavior of management and reducing opportunistic behaviors of management, such as earnings management and tax avoidance.

From the regression results in columns (2)-(3), the coefficient of information transparency (DACC) is significantly positive. This indicates a positive correlation between information transparency and corporate tax avoidance. Specifically, when a company's information transparency is low, it is more likely to engage in tax avoidance behavior. Companies with low information transparency have unclear financial statements and operating information, making it difficult for shareholders and other stakeholders to accurately understand the company's actual operation. This information asymmetry gives management more opportunities to engage in tax avoidance and other opportunistic behaviors without being easily detected. Therefore, companies with low information transparency are more likely to engage in tax avoidance to maximize the personal interests of management or the short-term interests of the company.

Column (4)-(6) are the regression results of agency costs (*agencycosts*) as the mediating variable. In column (4), analyst coverage is significantly negatively correlated with corporate agency costs at the 1% level, indicating that analyst coverage has effectively reduced corporate agency costs. From the regression results in column (5)-(6), the relationship between analyst coverage and corporate tax avoidance is still negative, and it is significant at the 1% level. Agency costs are positively correlated with corporate tax avoidance, and are significant at the 5% level, indicating that agency costs play a part in the mediating effect in analyst coverage to restrain corporate tax avoidance. Analysts' attention reduces the degree of information asymmetry between shareholders and management. This transparency makes management more considerate of the impact of their actions on shareholders' interests when making decisions, thereby reducing management's self-interested behavior. It curbs management's motivation to seek personal gains through concealed tax avoidance activities and engage in excessive on-the-job consumption, lowers the agency cost problem of enterprises,

**Table 5. Impact path test.**

| variable | (1) DACC | (2) BTD | (3) RATE1 | (4) Agencycost | (5) BTD | (6) RATE1 |
|---|---|---|---|---|---|---|
| Analyst | −0.002** | −0.004*** | −0.007*** | −0.043*** | −0.004*** | −0.007*** |
|  | (−2.07) | (−4.70) | (−4.03) | (−4.92) | (−4.67) | (−4.09) |
| DACC |  | 0.042*** | 0.093*** |  |  |  |
|  |  | (3.74) | (4.84) |  |  |  |
| Agencycost |  |  |  |  | 0.005** | 0.016*** |
|  |  |  |  |  | (2.22) | (3.45) |
| LnSize | −0.004*** | 0.003*** | 0.003** | 0.032*** | 0.003*** | 0.003* |
|  | (−5.54) | (3.12) | (1.85) | (3.13) | (3.03) | (1.88) |
| Lev | 0.020*** | −0.034*** | 0.066*** | −0.466*** | −0.033*** | 0.067*** |
|  | (4.11) | (−7.07) | (6.56) | (−8.60) | (−6.89) | (6.60) |
| Growth | 0.004** | −0.008*** | −0.006** | −0.037*** | −0.008*** | −0.005** |
|  | (2.36) | (−2.77) | (−2.33) | (−4.22) | (−2.65) | (−2.10) |
| PPE | −0.003 | 0.022*** | 0.089*** | 0.103* | 0.017*** | 0.078*** |
|  | (−0.44) | (4.40) | (7.65) | (1.73) | (3.38) | (6.82) |
| ROE | 0.083*** | 0.110*** | 0.166*** | −1.185*** | 0.120*** | 0.196*** |
|  | (7.69) | (7.14) | (7.56) | (−17.55) | (7.42) | (8.61) |
| ROI | 0.001*** | −0.000 | −0.003 | 0.004** | −0.000 | −0.000 |
|  | (2.89) | (−1.50) | (−0.80) | (2.05) | (−1.48) | (−0.84) |
| Intang | −0.066*** | −0.001 | 0.039 | 0.178* | −0.008 | 0.019 |
|  | (−4.71) | (−0.09) | (1.13) | (1.95) | (−0.60) | (0.56) |
| INV | 0.020** | 0.001 | −0.024 | −0.165*** | 0.005 | −0.018 |
|  | (2.12) | (0.19) | (−1.43) | (−3.34) | (0.65) | (−1.06) |
| _cons | 0.146*** | 0.007 | 0.075* | −1.038*** | 0.011 | 0.081** |
|  | (5.38) | (0.37) | (1.85) | (−3.99) | (0.53) | (2.00) |
| Industry | Yes | Yes | Yes | Yes | Yes | Yes |
| Year | Yes | Yes | Yes | Yes | Yes | Yes |
| Stock | Yes | Yes | Yes | Yes | Yes | Yes |
| N | 8,613 | 8,613 | 8,613 | 8,613 | 8,613 | 8,613 |
| R² | 0.106 | 0.106 | 0.193 | 0.417 | 0.104 | 0.186 |

Note: The P value in parentheses, *, **, *** indicate significant at the level of 10%, 5%, and 1%, respectively.

enhances the consistency of interests between management and shareholders, and makes management more inclined to make decisions from the perspective of maximizing enterprise value rather than for personal interests through tax avoidance. Therefore, analysts' attention indirectly reduces enterprises' tax avoidance behavior by lowering agency costs.

## 4.4. Robustness checks

**4.4.1. Variable replacement.** Accounting tax differences after removing the impact of accrued profits (*DDBTD*) and incorporating the difference between nominal tax rate and effective tax rate calculated without deducting deferred tax expense (*RATE2*) are used as the measure of corporate tax avoidance. The calculation method of *DDBTD* is shown in model (4) and model (5). *RATE2* equals income tax expense divided by EBIT.

$$BTD_{i,t} = \alpha TACC_{i,t} + \mu_i + \xi_{i,t} \tag{4}$$

$$DDBTD_{i,t} = \mu_i + \xi_{i,t} \tag{5}$$

*TACC* is the total accrued profit, which is equal to (net profit-net cash flow from operating activities)/total assets of the previous year; $\mu_i$ represents the mean value of the residual of company *i* during the sample period; $\xi_{i,t}$ represents the difference between residual of *t* year and the average residual $\mu_i$; *DDBTD* is the sum of $\mu_i$ and $\xi_{i,t}$, representing the part of the *BTD* that cannot be explained by the accrued profit.

From the regression results of columns (1) and (2) in Table 6, a significant negative correlation between analyst coverage and corporate tax avoidance remains.

**4.4.2. The explanatory variable lags by one period.** We regress the explanatory variables (*Analyst*) with a one-period lag to eliminate the possible reverse causality between analyst coverage and corporate tax avoidance. From the regression results of columns (3) and (4) in Table 6, there is still a significant negative correlation between analyst coverage and corporate tax avoidance at the 1% level.

**4.4.3. Endogeneity test.** We use the 2SLS method to test the endogenous problem. Drawing on the research of [42], we select whether the company belongs to the Shanghai and Shenzhen 300 constituent stocks and the industry average number of analysts as instrumental variables. The CSI 300 constituent stocks are selected because the selection criteria for these relate to the company's influence in its industry, the number of transactions, and so on when the company is selected as a constituent stock. This is important for brokers and fund companies, and attracts the attention and follow-up of analysts, but does not affect the company's tax avoidance behavior. This makes the CSI 300 constituent stock an

**Table 6. Robustness test results.**

| Variable | (1) | (2) | Variable | (3) | (4) |
|---|---|---|---|---|---|
| | DDBTD | RATE2 | | BTD | RATE1 |
| Analyst | −0.004*** | −0.006*** | L1Analyst | −0.003*** | −0.007*** |
| | (−4.72) | (−3.93) | | (−2.91) | (−3.38) |
| ROE | 0.093*** | 0.056*** | L1ROE | 0.065*** | 0.098*** |
| | (6.13) | (3.04) | | (4.67) | (4.08) |
| ROI | −0.000* | 0.000 | L1ROI | −0.001* | −0.001* |
| | (−1.77) | (0.27) | | (−1.74) | (−1.86) |
| Growth | −0.009*** | 0.002 | L1Growth | −0.001 | 0.004 |
| | (−3.07) | (0.84) | | (−0.68) | (1.47) |
| PPE | 0.048*** | 0.045*** | L1PPE | 0.012** | 0.060*** |
| | (9.29) | (4.49) | | (1.99) | (4.42) |
| Intang | 0.025* | 0.001 | L1Intang | −0.004 | 0.005 |
| | (1.81) | (0.04) | | (−0.26) | (0.13) |
| Lev | −0.027*** | 0.088*** | L1Lev | −0.027*** | 0.054*** |
| | (−5.62) | (9.92) | | (−5.13) | (4.60) |
| INV | −0.010 | −0.023 | L1INV | 0.010 | −0.013 |
| | (−1.31) | (−1.61) | | (1.13) | (−0.65) |
| LnSize | 0.003*** | 0.003** | L1LnSize | 0.003*** | 0.004* |
| | (3.34) | (2.19) | | (2.64) | (1.84) |
| cons | −0.019 | 0.087** | _cons | 0.002 | 0.025 |
| | (−0.97) | (2.54) | | (0.06) | (0.52) |
| Industry | Yes | Yes | Industry | Yes | Yes |
| Year | Yes | Yes | Year | Yes | Yes |
| Stock | Yes | Yes | Stock | Yes | Yes |
| N | 8,613 | 8,613 | N | 5,910 | 5,910 |
| R² | 0.083 | 0.196 | R² | 0.075 | 0.170 |

Note: The P value in parentheses, *, **, *** indicate significant at the level of 10%, 5%, and 1%, respectively

ideal instrumental variable. In addition, when there are many analysts tracked by a certain industry, other analysts will be attracted to follow up, and the number of analysts tracked by other companies in the industry will not affect the company's tax avoidance behavior. This industry-wide analyst tracking effect further strengthens the rationality of the CSI 300 constituents as instrumental variables. Specifically, the selection criteria of CSI 300 constituents and the analyst tracking effect in the industry work together, so that the relationship between the number of analysts and corporate tax avoidance behavior can be effectively separated and identified through the instrumental variable of CSI 300 constituents.Table 7 represents the 2SLS regression results. Whether a company is selected as the CSI 300 component stock is significantly positively correlated with the number of analysts. Adding the predicted value of the first phase to the second phase of regression shows that the number of analysts is still related to corporate tax avoidance behavior. There is a significant

**Table 7. Endogenous test results (2SLS).**

| Variable | Instrumental variable – CSI300 component stocks | | | Instrumental variable – Industry average number of analysts | |
|---|---|---|---|---|---|
| | (1) | (2) | (3) | (4) | (5) |
| | Analyst | BTD | RATE1 | BTD | RATE1 |
| CSI300 | 0.144*** | | | | |
| | (4.07) | | | | |
| Analyst_hat | | −0.036** | −0.067** | | |
| | | (−2.20) | (−2.12) | | |
| Analyst-mean | | | | −0.021*** | −0.035*** |
| | | | | (−3.62) | (−2.89) |
| ROE | 4.131*** | 0.247*** | 0.429*** | 0.179*** | 0.311*** |
| | (25.73) | (3.67) | (3.18) | (7.31) | (6.47) |
| ROI | −0.016*** | −0.001** | −0.001* | −0.001** | −0.001 |
| | (−3.35) | (−2.49) | (−1.95) | (−2.15) | (−1.60) |
| Growth | −0.033 | −0.009*** | −0.008*** | −0.010*** | −0.011*** |
| | (−1.63) | (−3.03) | (−2.70) | (−3.21) | (−4.26) |
| PPE | −0.097 | 0.013*** | 0.070*** | 0.010** | 0.091*** |
| | (−1.02) | (2.60) | (5.98) | (1.97) | (8.26) |
| Intang | 0.416 | 0.006 | 0.050 | −0.019 | 0.025 |
| | (1.52) | (0.36) | (1.34) | (−1.26) | (0.77) |
| Lev | −1.060*** | −0.071*** | −0.005 | −0.052*** | 0.026 |
| | (−12.90) | (−3.84) | (−0.12) | (−6.50) | (1.58) |
| INV | −0.000 | 0.004 | −0.019 | −0.018*** | −0.064*** |
| | (−0.00) | (0.53) | (−1.09) | (−2.81) | (−4.41) |
| LnSize | 0.314*** | 0.014** | 0.024** | 0.007*** | 0.012*** |
| | (20.36) | (2.41) | (2.20) | (3.65) | (2.92) |
| cons | −4.442*** | −0.155* | −0.233 | −0.096*** | −0.211*** |
| | (−10.67) | (−1.84) | (−1.42) | (−3.38) | (−3.35) |
| Year | Yes | Yes | Yes | Yes | Yes |
| Industry | Yes | Yes | Yes | No | No |
| Stock | Yes | Yes | Yes | Yes | Yes |
| N | 8,613 | 8,613 | 8,613 | 8,613 | 8,613 |
| $R^2$/chi2 | 0.380 | 0.099 | 0.180 | 193.829 | 439.644 |

Note: The P value in parentheses, *, **, *** indicate significant at the level of 10%, 5%, and 1%, respectively.

 

negative correlation, which confirms the robustness of our findings. The regression results using the industry average number of analysts (*Analyst-mean*) as an instrumental variable are also consistent.

## 5. Further analysis

### 5.1. The impact of innovation investment

At present, China vigorously promotes mass entrepreneurship and innovation, and innovation activity has increased. However, the risk of innovation is higher, because in entails more investment, the results of which is uncertain. This puts pressure on profitability expectations, and in turn on management [43], but may also improve performance. If operating results do not reach the expected level, the capital chain is at risk, putting greater pressure on management's profitability. However, innovation activities need sufficient and stable cash flow to ensure their sustainability. Han and Yan [44] find that corporate innovation activities are more dependent on external financing, and this dependence is more obvious in small and medium-sized enterprises, private enterprises, and high-tech enterprises. Since China's financial system is dominated by indirect financing, the banking market structure is monopolized by state-owned banks, and bank loans emphasize physical assets over intangible assets, which leads to a large funding bottleneck for corporate innovation activities. Therefore, when external financing is limited, managers have a stronger incentive to save cash through tax avoidance. At this time, the "supervisory role" of analyst coverage may have a greater impact on companies with high investment in innovation, and analysts' attention to the inhibiting effect of corporate tax avoidance is weakened.

The ratio of R&D expenses to operating income is selected as a proxy variable for enterprise innovation input (R&D), and the interaction term of analyst coverage and innovation input is added to model (1) to construct model (6), which mainly focuses on the interaction term $\alpha_2$, It is expected that $\alpha_2$ is significantly positive.

$$TA_{i,t} = \alpha_0 + \alpha_1 Analyst_{i,t} + \alpha_2 Analyst\_R\&D + \alpha_3 R\&D + \alpha_4 ROE_{i,t} + \alpha_5 ROI_{i,t} + \alpha_6 Growth_{i,t}$$
$$+ \alpha_7 PPE_{i,t} + \alpha_8 Intang_{i,t} + \alpha_9 Lev_{i,t} + \alpha_{10} INV_{i,t} + \alpha_{11} LnSize_{i,t} + Year + Indcd + Stock + \xi_{i,t} \tag{6}$$

The regression results of model (6) are shown in Table 8. Analyst coverage is significantly negatively correlated to corporate tax avoidance, and the crossover item between analyst coverage and corporate innovation input is significantly positive, indicating that corporate innovation input has weakened the inhibiting effect of analyst coverage on corporate tax avoidance. The reason is that when enterprises invest more in innovation, the profitability pressure and capital demand faced by management are increased, making them more inclined to obtain funds to support innovation activities through tax avoidance. At this time, although the attention of analysts will still have a certain supervisory effect on corporate tax avoidance, the effect of this supervision will be weakened to a certain extent due to the strong tax avoidance motivation of the management.

### 5.2. The impact of nature of property rights

Tax payment in accordance with the law is the obligation and responsibility of all enterprises, but the nature of property rights of Chinese enterprises is clearly distinguished. State-owned enterprises have high agency costs due to the absence of owners [45]. Ping, Fan, and Hao [46] find that the agency cost of Chinese state-owned enterprises is equivalent to 60%–70% of the profit potential, and the agency cost effects the company's efficiency only 30%–40%. So, does analyst coverage have differing impacts on companies with different property rights? In this paper, the samples are grouped according to the nature of property rights, and the regression results are shown in Table 9. The inhibiting effect of analysts' concerns on tax avoidance is significant in both state-owned and non-state-owned enterprises. When *BTD* is used as a measurement indicator, there is no significant difference in the coefficients between the two groups. However, when *RATE1* is used, there is a significant difference in the coefficient between the two groups at the 1% level, and it can be seen from the coefficient that the coefficient in the state-owned enterprise group is higher, indicating that in the absence of

**Table 8. Further regression results (1).**

| Variable | (1) | (2) |
|---|---|---|
| | BTD | RATE1 |
| Analyst | −0.006*** | −0.012*** |
| | (−5.34) | (−5.17) |
| R&D | −0.001** | −0.004*** |
| | (−2.53) | (−3.77) |
| Analyst_R&D | 0.001*** | 0.002*** |
| | (2.69) | (3.37) |
| ROE | 0.115*** | 0.175*** |
| | (7.45) | (8.04) |
| ROI | −0.000 | −0.000 |
| | (−1.31) | (−0.53) |
| Growth | −0.008*** | −0.006** |
| | (−2.68) | (−2.15) |
| PPE | 0.018*** | 0.077*** |
| | (3.46) | (6.64) |
| Intang | −0.007 | 0.027 |
| | (−0.52) | (0.78) |
| Lev | −0.035*** | 0.060*** |
| | (−7.23) | (5.87) |
| INV | 0.005 | −0.019 |
| | (0.65) | (−1.11) |
| LnSize | 0.003*** | 0.004* |
| | (3.30) | (1.95) |
| cons | 0.006 | 0.081** |
| | (0.31) | (1.99) |
| Year | Yes | Yes |
| Industry | Yes | Yes |
| Stock | Yes | Yes |
| N | 8,613 | 8,613 |
| $R^2$ | 0.105 | 0.185 |

Note: The P value in parentheses, *, **, *** indicate significant at the level of 10%, 5%, and 1%, respectively.

state-owned enterprise owners in China, the agency cost is higher, and the supervisory role played by analyst coverage is more significant. The reason is that the alignment of interests between the management of SOEs and shareholders (the state) is weaker, and the management has a stronger self-interest motive, and is more likely to seek private interests through tax evasion and other behaviors. However, analyst attention can improve the information transparency of enterprises, reduce the degree of information asymmetry, so as to more effectively supervise the behavior of the management layer and inhibit their tax avoidance motives, so the inhibitory effect of analyst attention is more prominent in state-owned enterprises.

### 5.3. The impact of equity incentives

According to the optimal contract theory, equity is granted to a firm's management. Equity incentives can coordinate the private interests of management and shareholders, prompting corporate managers to make decisions from the

**Table 9. Further regression results (2).**

| Variable | State-owned | | Non-state | | High shareholding group | | Low shareholding group | |
|---|---|---|---|---|---|---|---|---|
| | (1) | (2) | (3) | (4) | (5) | (6) | (7) | (8) |
| | BTD | RATE1 | BTD | RATE1 | BTD | RATE1 | BTD | RATE1 |
| Analyst | −0.004*** | −0.011*** | −0.003*** | −0.003* | −0.002* | −0.002 | −0.005*** | −0.009*** |
| | (−4.54) | (−4.97) | (−3.71) | (−1.72) | (−1.70) | (−0.86) | (−5.36) | (−4.70) |
| ROE | 0.093*** | 0.166*** | 0.124*** | 0.176*** | 0.131*** | 0.219*** | 0.101*** | 0.144*** |
| | (9.64) | (7.31) | (12.47) | (9.51) | (11.80) | (10.20) | (11.18) | (7.37) |
| ROI | −0.001** | −0.002* | −0.000 | 0.000 | −0.000 | −0.001 | −0.000 | 0.000 |
| | (−2.34) | (−1.76) | (−0.27) | (0.61) | (−1.43) | (−1.08) | (−0.44) | (0.38) |
| Growth | −0.008*** | −0.007* | −0.008*** | −0.005** | −0.013*** | −0.005 | −0.005*** | −0.006** |
| | (−5.39) | (−1.95) | (−6.04) | (−1.98) | (−8.08) | (−1.50) | (−4.45) | (−2.38) |
| PPE | 0.012** | 0.059*** | 0.021*** | 0.088*** | 0.022*** | 0.093*** | 0.009 | 0.054*** |
| | (2.19) | (4.67) | (3.58) | (8.12) | (3.64) | (7.95) | (1.61) | (4.73) |
| Intang | −0.013 | 0.027 | −0.006 | 0.006 | −0.006 | 0.048 | −0.021 | −0.014 |
| | (−0.93) | (0.83) | (−0.35) | (0.18) | (−0.30) | (1.25) | (−1.56) | (−0.46) |
| Lev | −0.039*** | 0.034*** | −0.037*** | 0.081*** | −0.034*** | 0.064*** | −0.041*** | 0.052*** |
| | (−8.63) | (3.19) | (−7.90) | (9.23) | (−7.10) | (6.82) | (−9.04) | (5.31) |
| INV | 0.001 | −0.023 | 0.016** | −0.004 | 0.025*** | 0.026* | −0.009 | −0.051*** |
| | (0.12) | (−1.43) | (2.31) | (−0.32) | (3.23) | (1.71) | (−1.33) | (−3.61) |
| LnSize | 0.001* | 0.003* | 0.003*** | 0.004** | 0.001 | 0.001 | 0.002*** | 0.003* |
| | (1.92) | (1.83) | (3.80) | (2.31) | (1.17) | (0.43) | (3.08) | (1.75) |
| cons | 0.023 | 0.120*** | 0.004 | 0.023 | 0.022 | 0.014 | 0.029 | 0.149*** |
| | (1.25) | (2.76) | (0.19) | (0.56) | (0.94) | (0.31) | (1.59) | (3.84) |
| Industry | Yes | Yes | Yes | Yes | Yes | Yes | Yes | Yes |
| Year | Yes | Yes | Yes | Yes | Yes | Yes | Yes | Yes |
| Stock | Yes | Yes | Yes | Yes | Yes | Yes | Yes | Yes |
| N | 3,784 | 3,784 | 4,829 | 4,829 | 4,159 | 4,159 | 4,454 | 4,454 |
| F | 9.17*** | 14.00*** | 7.60*** | 11.90*** | 6.98*** | 9.27*** | 7.61*** | 13.61*** |
| R² | 0.146 | 0.213 | 0.103 | 0.159 | 0.103 | 0.137 | 0.117 | 0.201 |

Note: The P value in parentheses, *, **, *** indicate significant at the level of 10%, 5%, and 1%, respectively.

perspective of maximizing the value of the company [47]. When the management shareholding ratio is high, it will be more consistent with shareholders' interests, prioritizing corporate value in decision-making and exercising caution in regard to tax avoidance behavior. Therefore, the impact of analysts' concerns on corporate tax avoidance will be more pronounced in companies with low management holdings. Using the median of the sample of management's shareholding ratio, the sample is divided into the higher shareholding ratio group and the lower shareholding ratio group. The regression results are shown in Table 9. Whether *BTD* or *RATE1* is used as a measure of corporate tax avoidance, the coefficient of the higher management shareholding group is higher than that of the group with a lower management shareholding ratio, and the coefficient difference between the groups is significant at 5% and 1% respectively. It shows that analysts pay more attention to the stronger "supervisory effect" in companies with a lower management shareholding ratio. This is because when management's shareholding ratio is high, its interests are highly aligned with shareholders, and the agency cost is relatively low. Where there is a lower management shareholding ratio, the interests are not aligned and management is more self-interested, bringing higher agency costs. Then, analyst coverage will exert a stronger inhibiting effect.

## 6. Conclusions

This article uses data from Chinese A-share listed companies for the period 2009–2021 to empirically test the impact of analysts' concerns on corporate tax avoidance behavior. The study found that analysts' attention will significantly inhibit corporate tax avoidance behavior, playing a "supervisory effect". Also, by testing the mechanisms underlying this, we found that the analyst's concerns on corporate tax avoidance behavior are mainly through improving the information environment and reducing corporate agency costs. When the information environment of the enterprise is poor or the agency cost is high, managements and shareholders' interests are not consistent, and management is more likely to seek personal gain through tax avoidance and other behaviors. Analysts are concerned about increasing the probability of failure of management's tax avoidance behavior and the cost to be paid, and by weighing the costs and benefits, management will reduce tax avoidance behavior. Further analysis finds that the role of analysts in restraining corporate tax avoidance in companies that invest more in innovation is weakened. In a sample with a low proportion of state-owned enterprises and management holdings, analysts exert a stronger restraining effect.

Tax avoidance behavior intensifies the degree of corporate information asymmetry. When external supervision mechanisms such as analysts exist, they can reduce corporate information asymmetry, thereby reducing corporate agency costs, inhibiting management's motivation to avoid taxation, and allowing corporate governance to be effective. Based on our findings, we identify the following recommendations for policymakers. First, regulators should pay attention to the role played by analysts and other intermediary agencies. The Chinese investor protection system is in its infancy, and external governance mechanisms such as analysts can play a part in restraining the self-interested behavior of management. However, formulate special regulations to clarify the code of conduct and independence requirements for analysts in emerging markets, prohibit the transfer of interests between analysts and listed companies, such as prohibiting analysts from accepting gifts and travel invitations from listed companies, and regulators should regularly assess the independence of analysts, and check whether analysts have conflicts of interest through spot checks on research reports and investor feedback. Prevent analysts from losing their independence and objectivity as an external governance mechanism due to certain interests, resulting in chaos and instability in the financial market. We find that information environment and agency problem are some factors that affect tax avoidance,

Tax regulation cannot curb tax avoidance by all businesses. Therefore, in addition to daily law enforcement activities such as inspections and inspections, the tax regulatory authorities can use the relevant tax-related information provided by analysts to conduct tax law enforcement inspections, to manage the misconduct of enterprises in a more targeted manner and reduce the loss of national tax revenue. For state-owned enterprises, improving the corporate governance structure is the key to enhancing corporate value and reducing tax evasion. Improve the level of automatic control, and realize the standardization, process and intelligence of financial internal control. In terms of tax management, state-owned enterprises should improve the tax support mechanism for major decision-making, understand the characteristics of their business and study the major issues and major problems existing in the process of daily tax management through the analysis and collection of business needs, so as to further improve the tax management system.

## Author contributions

**Conceptualization:** Yuanfang Wang.

**Investigation:** Jinlong Han.

**Writing – original draft:** Yuanhao Shen.

**Writing – review & editing:** Xiaofei Shi.

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
