## [Decision Letter · Decision Letter 0]

16 Dec 2024

PONE-D-24-37181Can Analyst Coverage reduce Corporate Tax Avoidance? Evidence from ChinaPLOS ONE?

Dear Dr. Shi,

Thank you for submitting your manuscript to PLOS ONE. After careful consideration, we feel that it has merit but does not fully meet PLOS ONE’s publication criteria as it currently stands. 

In view of the referees’ feedback and my own reading of your paper, we invite you to address all issues noted below. We consider these issues to be major in nature, requiring more than a superficial or minor revision. We have particular concerns about methodological part of the manuscript. We consider that the authors should make an important effort improving this section.

Since our point of view the paper has an important potential to be consider for publication on this journal. We are confident that the issues identified could be resolved with a major revision, so we invite you to address the issues noted below and resubmit the manuscript for a new revision round. 

We look forward to receiving your revised manuscript.

Kind regards,

Juan E. Trinidad-Segovia, PhD

Section Editor

PLOS ONE

Journal Requirements:

This work was supported by the Key Project of Social Science Research of Beijing Municipal Education Commission (SZ202210038021/21GJB021), New Finance and Economics Research Project of Hebei University of Business and Economics (2020XCJ03)。  

Reviewers' comments:

Reviewer's Responses to Questions

**Comments to the Author**

1. Is the manuscript technically sound, and do the data support the conclusions?

Reviewer #1: Yes

Reviewer #2: Yes

2. Has the statistical analysis been performed appropriately and rigorously?

Reviewer #1: Yes

Reviewer #2: No

3. Have the authors made all data underlying the findings in their manuscript fully available?

Reviewer #1: Yes

Reviewer #2: No

4. Is the manuscript presented in an intelligible fashion and written in standard English?

Reviewer #1: No

Reviewer #2: Yes

Reviewer #1: Your manuscript examining analyst coverage and corporate tax avoidance presents valuable insights but requires minor revisions to enhance its contribution:

The theoretical framework requires deeper engagement with institutional theory. While you effectively present competing hypotheses regarding analyst impact, the theoretical underpinning needs stronger connection to emerging market contexts. Specifically, elaborate on how China's institutional environment affects analyst independence and behavior differently from developed markets. The state ownership context deserves particular attention given its significance in your findings.

Your empirical approach demonstrates considerable rigor, but several aspects need enhancement. First, while your identification strategy using CSI 300 inclusion is clever, more discussion of the exclusion restriction would strengthen the argument. Second, the path analysis results would benefit from clearer presentation - consider adding a path diagram illustrating the direct and indirect effects through information environment and agency costs.

The moderation analysis reveals important contingencies, but requires stronger theoretical development. The finding that innovation investment weakens analyst influence is intriguing but needs more thorough explanation. Similarly, the state ownership results warrant deeper discussion of how ownership structure interacts with analyst monitoring effectiveness.

Your policy implications could be more specific. Consider developing concrete recommendations for:

Regulatory oversight of analyst independence in emerging markets

Integration of analyst research into tax enforcement mechanisms

Governance requirements for state-owned enterprises

These revisions will enhance your paper's contribution to both corporate governance and tax avoidance literature. The findings have particular relevance for emerging markets developing their institutional frameworks for corporate monitoring and tax compliance.

Your research makes important strides in understanding external monitoring mechanisms. These suggested revisions will strengthen its impact and applicability for both scholars and policymakers.

Reviewer #2: This paper studies the impact of analysts coverage on tax avoidance behavior, proving that it significantly inhibits this behavior by improving the information environment and alleviating agency problems. I find this article interesing, but there are some aspects that need to be improved in order it to be publishable:

1. On section 2.2. authors should add papers from each of the presented views.

2. On Figure 1, the hypothesis being tested should be represented on the figure.

3. Which is the periodicity of data? Daily data? Monthly data? Please, say it.

4. You should present all variables, one by one. Are there control variables?

5. Where is the methodology section?

6. How data has been treated? How the final sample has been constructed? What about outliers?

7. You should say something more about the data presented on Table 2.

8. Before presenting the results, you should say something about the existence (or not) of multicollinearity between variables being tested.

9. Table 4 should be changed to improve reader's understanding.

10. Finally, I find you should add a discussion section.

**Do you want your identity to be public for this peer review?** For information about this choice, including consent withdrawal, please see our Privacy Policy

Reviewer #1: No

Reviewer #2: No

---

## [Author Response · Author response to Decision Letter 1]

16 Feb 2025

We feel great thanks for your professional review work on our article. As you are concerned, there are several problems that need to be addressed. According to your nice suggestions, we have made extensive corrections to our previous draft, the detailed corrections are listed below.

Reviewer #1:

1、The theoretical framework requires deeper engagement with institutional theory. While you effectively present competing hypotheses regarding analyst impact, the theoretical underpinning needs stronger connection to emerging market contexts. Specifically, elaborate on how China's institutional environment affects analyst independence and behavior differently from developed markets. The state ownership context deserves particular attention given its significance in your findings.

The author's answer: Compared with foreign developed markets, the focus of Chinese analysts is more complex. In developed markets, where corporate governance mechanisms are more mature and investor protection laws are well-established, analysts are likely to play a more intermediary role in influencing market expectations and stock prices by providing professional analysis. In China, analysts not only act as information intermediaries, but also assume the role of external governance mechanisms to a certain extent, and their oversight effect plays an important role in curbing corporate tax evasion and other misconduct. At the same time, the peculiarities of the Chinese market also make analysts' optimistic forecasts and other behaviors may exert more pressure on management and affect their decision-making. Therefore, the influence mechanism and effect that analysts focus on in China's institutional environment are significantly different from those in foreign developed markets.

2、Your empirical approach demonstrates considerable rigor, but several aspects need enhancement. First, while your identification strategy using CSI 300 inclusion is clever, more discussion of the exclusion restriction would strengthen the argument. Second, the path analysis results would benefit from clearer presentation - consider adding a path diagram illustrating the direct and indirect effects through information environment and agency costs.

The author's answer: When the company is selected as a constituent stock, it becomes more important for brokerages and fund companies, and it will attract more attention and tracking from analysts, but whether it is a CSI 300 constituent stock will not affect the company's tax avoidance behavior. Therefore, whether it belongs to the CSI 300 constituent stocks is related to the number of analysts who pay attention, but it will not have a direct impact on corporate tax avoidance. This makes the CSI 300 constituent stock an ideal instrumental variable. In addition, when there are more analysts in a certain industry, other analysts will be attracted to follow, and the number of analysts in other companies in the industry will not affect the company's tax avoidance behavior. This industry-wide analyst tracking effect further strengthens the rationality of the CSI 300 constituents as instrumental variables. Specifically, the selection criteria of CSI 300 constituents and the analyst tracking effect in the industry work together, so that the relationship between the number of analysts and corporate tax avoidance behavior can be effectively separated and identified through the instrumental variable of CSI 300 constituents.

Supplement to the mediator test effect:

Table 5 presents the regression results of models (2) and (3). Columns (1)-(3) show the regression results with information transparency (DACC) as the mediating variable. In column (1), the coefficient of analyst attention (Analyst) and information transparency (DACC) is negative and significant. This indicates a significant negative correlation between analyst attention and information transparency. Specifically, when the attention of analysts towards a company increases, the company's information transparency significantly improves. Analysts, by continuously monitoring the company and releasing research reports, convey detailed information about the company to the market, including financial status, operating performance, and future prospects. This information dissemination makes the company's information more transparent and reduces the information asymmetry between shareholders and management. The improvement in information transparency means that shareholders can better understand the actual operation of the company, thereby better supervising the behavior of management and reducing opportunistic behaviors of management, such as earnings management and tax avoidance.

From the regression results in columns (2)-(3), it can be seen that the coefficient of information transparency (DACC) is significantly positive. This indicates a positive correlation between information transparency and corporate tax avoidance. Specifically, when a company's information transparency is low, it is more likely to engage in tax avoidance behavior. Companies with low information transparency have unclear financial statements and operating information, making it difficult for shareholders and other stakeholders to accurately understand the company's actual operation. This information asymmetry gives management more opportunities to engage in tax avoidance and other opportunistic behaviors without being easily detected. Therefore, companies with low information transparency are more likely to engage in tax avoidance to maximize the personal interests of management or the short-term interests of the company.

3、The moderation analysis reveals important contingencies, but requires stronger theoretical development. The finding that innovation investment weakens analyst influence is intriguing but needs more thorough explanation. Similarly, the state ownership results warrant deeper discussion of how ownership structure interacts with analyst monitoring effectiveness.

The author's answer:

(1)R&D:The regression results for model (6) are shown in Table 8. There is a significant negative correlation between analysts' attention and corporate tax avoidance, and the intersection term between analysts' attention and corporate innovation investment is significantly positive, indicating that corporate innovation investment weakens the inhibitory effect of analysts' attention on corporate tax avoidance. The reason is that when enterprises invest more in innovation, the profitability pressure and capital demand faced by management are increased, making them more inclined to obtain funds to support innovation activities through tax avoidance. At this time, although the attention of analysts will still have a certain supervisory effect on corporate tax avoidance, the effect of this supervision will be weakened to a certain extent due to the strong tax avoidance motivation of the management.

(2)Nature of property rights:The regression results are shown in Table 9 below. The inhibitory effect of analysts' attention on tax avoidance is significant in both state-owned and non-state-owned enterprises. When BTD was used as a measure, there was no significant difference in the coefficient between the two groups. However, when RATE1 is used as the measurement index, there is a significant difference in the coefficient between the two groups at the 1% level, and it can be seen from the coefficient that the coefficient is higher in the state-owned enterprise group, indicating that in the absence of owners of state-owned enterprises in China, the agency cost is higher, and the supervisory role played by analysts' attention is more significant. The reason is that the alignment of interests between the management of SOEs and shareholders (the state) is weaker, and the management has a stronger self-interest motive, and is more likely to seek private interests through tax evasion and other behaviors. However, analyst attention can improve the information transparency of enterprises, reduce the degree of information asymmetry, to more effectively supervise the behavior of the management layer and inhibit their tax avoidance motives, so the inhibitory effect of analyst attention is more prominent in state-owned enterprises.

4、Your policy implications could be more specific. Consider developing concrete recommendations for:

Regulatory oversight of analyst independence in emerging markets

Integration of analyst research into tax enforcement mechanisms

Governance requirements for state-owned enterprises

The author's answer:

Based on the conclusions of the study, the following suggestions are put forward: (1) The regulatory authorities should pay attention to the role played by intermediaries such as analysts. China's investor protection system is still immature, and external governance mechanisms such as analysts can exert some supervisory effects and inhibit the self-interested behavior of management. However, at present, this kind of external governance mechanism is not the leading force in China's capital market, and it is necessary to pay more attention to it so that it can better play the role of corporate governance. Formulate special regulations to clarify the code of conduct and independence requirements for analysts in emerging markets, prohibit the transfer of interests between analysts and listed companies, such as prohibiting analysts from accepting gifts and travel invitations from listed companies, and regulators should regularly assess the independence of analysts, and check whether analysts have conflicts of interest through spot checks on research reports and investor feedback. Prevent analysts from losing their independence and objectivity as an external governance mechanism due to certain interests, resulting in chaos and instability in the financial market. (2) Through the intermediary effect test, this paper finds that the information environment and agency problem are the factors that cause corporate tax avoidance, and this problem is also a common problem of enterprises, so the governance level of enterprises should pay attention to and take measures to improve the information transparency of enterprises and reduce agency costs, such as establishing reasonable executive incentives, moderately increasing the proportion of management shareholdings, improving the sensitivity of their compensation performance, reducing corporate tax avoidance behavior, and enhancing corporate value. Tax evasion is widespread, and due to the constraints of resource conditions and cost factors, tax regulation cannot curb all corporate tax avoidance. Therefore, in addition to daily law enforcement activities such as inspections and inspections, the tax regulatory authorities can use the relevant tax-related information provided by analysts to conduct tax law enforcement inspections, so as to manage the misconduct of enterprises in a more targeted manner and reduce the loss of national tax revenue. (3) For state-owned enterprises, improving the corporate governance structure is the key to enhancing corporate value and reducing tax evasion. State-owned enterprises should further deepen the classification reform, classification assessment, classification accounting, improve the classification assessment and evaluation system of state-owned enterprises, and set up more targeted and personalized assessment indicators according to the different functions of enterprises. At the same time, it is necessary to actively promote precise management and control and implement different management and control modes such as strategic, operational, and governance according to the equity structure, management level, and role positioning of the subordinate units. In addition, state-owned enterprises need to improve a comprehensive and effective compliance and risk control system, establish, and improve the financial internal control system, refine the control measures of key links, improve the level of automatic control, and realize the standardization, process and intelligence of financial internal control. In terms of tax management, state-owned enterprises should improve the tax support mechanism for major decision-making, understand the characteristics of their business and study the major issues and major problems existing in the process of daily tax management through the analysis and collection of business needs, to further improve the tax management system.

Reviewer #2:

1、On section 2.2. authors should add papers from each of the presented views.

The author's answer:

References have been indicated in the content discussed in Section 2.2.

Two opposing views exist on the impact of tax avoidance on enterprises. The traditional view considers that there is no agency cost between shareholders and management, and tax avoidance supports a company's cash flow, providing more cash for management to invest in projects that increase the value of the company, in turn having a positive impact on the company(Zhan and Liu 2021). However, the opposing view based on principal‒agent theory posits that the interests of shareholders and management are inconsistent, and managers are more likely to put their own interests above those of shareholders and the company. Tax avoidance behavior usually requires complex planning, which strengthens the degree of information asymmetry between shareholders and management. Managers’ self-interest means that they are more likely to use their information advantage to invest cash resulting from tax avoidance for on-the-job consumption(Cheng et al 2016).

However, there is no consensus in the literature as to the impact of analysts on corporate tax avoidance behavior(Chen and Xu 2015). First, some scholars believe that analysts' coverage will reduce corporate tax avoidance because when analysts pay attention to a company, their evaluations are published in research reports that help investors better understand the company and reduce the degree of information asymmetry(Bradley et al,2017). Moreover, because analysts have rich professional knowledge and industry experience, they have the expertise to discover earnings management behaviors such as corporate tax avoidance activities (Daniel 2017), so analyst coverage has a "supervisory effect". Second, investors have a high degree of trust in analysts’ research reports (Tivoli and Lakonishok 1979). If the research report includes negative news about corporate tax avoidance behavior investors’ expectations of the company are lowered, leading to a decline in stock prices and damaging the company's value and reputation (Chen and Wu 2019). It will also attract the attention of tax regulatory authorities, and possible penalties may ensue. Therefore, when managers plan tax avoidance activities, they must consider the costs and benefits. Doing so suggests the following hypothesis:

H1a: When analysts’ concerns increase, the degree of corporate tax avoidance decreases.

2、On Figure 1, the hypothesis being tested should be represented on the figure.

The author's answer:

H1a and H1b are already represented on the diagram, as detailed in the diagram below.

Figure 1 Logical framework.

3、Which is the periodicity of data? Daily data? Monthly data? Please, say it.

The author's answer: This article uses annual data.

4、 You should present all variables, one by one. Are there control variables?

The author's answer: A description of the types of variables, such as explanatory variables, explanatory variables, control variables, etc., has been added to the text.

Type Variable Description Definition

Dependent variable BTD Accounting ‒ tax differences The difference between total accounting profit and taxable income. The calculation formula is: tax difference = (accounting profit before tax ‒ taxable income)/total assets of the previous year; Taxable income = (income tax expense ‒ deferred income tax expense)/nominal income tax rate

RATE1 Nominal tax rate ‒ effective tax rate difference (Income tax expense ‒ deferred income tax expense)/EBIT

Argument Analyst Analyst coverage Add one to the number of analysts tracking and take the natural logarithm

Mediation variables Agencycost Agency costs Operating income for the period) / total assets at the end of the period

Adjust variables R&D Invest in innovation R&

---

## [Decision Letter · Decision Letter 1]

2 Mar 2025

Can Analyst Coverage reduce Corporate Tax Avoidance? Evidence from China

PONE-D-24-37181R1

Dear Dr. Wang,

We’re pleased to inform you that your manuscript has been judged scientifically suitable for publication and will be formally accepted for publication once it meets all outstanding technical requirements.

Kind regards,

Juan E. Trinidad-Segovia, PhD

Section Editor

PLOS ONE

Additional Editor Comments (optional):

Reviewers' comments:

Reviewer's Responses to Questions

**Comments to the Author**

Reviewer #2: All comments have been addressed

2. Is the manuscript technically sound, and do the data support the conclusions?

Reviewer #2: Yes

3. Has the statistical analysis been performed appropriately and rigorously?

Reviewer #2: Yes

4. Have the authors made all data underlying the findings in their manuscript fully available?

Reviewer #2: Yes

5. Is the manuscript presented in an intelligible fashion and written in standard English?

Reviewer #2: Yes

Reviewer #2: All comments made have been correctly addressed. For me the paper is now ready to be published. Thank you for the work made.

**Do you want your identity to be public for this peer review?** For information about this choice, including consent withdrawal, please see our Privacy Policy

Reviewer #2: No

---

## [Editor Report · Acceptance letter]

PONE-D-24-37181R1

PLOS ONE

Dear Dr. Wang,

I'm pleased to inform you that your manuscript has been deemed suitable for publication in PLOS ONE. Congratulations! Your manuscript is now being handed over to our production team.

Kind regards,

on behalf of

Dr. Juan E. Trinidad-Segovia

Section Editor

PLOS ONE